# OpenReview forum: "Beyond Self‑Attention: A Subquadratic Fourier‑Wavelet Transformer with Multi‑Modal Fusion"
_NeurIPS.cc/2025/Conference — Submitted to NeurIPS 2025_

### Official Review · Reviewer_Zn3D · 2025-06-29

**Clarity:** 3
**Significance:** 2
**Originality:** 2
**Rating:** 3
**Confidence:** 4

**Summary:**

This paper introduces a modified efficient transformer architecture called the Multi-Domain Fourier-Wavelet Attention (MDFWA) Transformer, designed to address the quadratic computational complexity of traditional self-attention mechanisms in transformers. In particular, MDFWA replaces self-attention with Fourier spectral techniques, especially, Fourier Transform-based token mixing combined with localized wavelet filtering. This approach integrates global Fourier token mixing with discrete wavelet transforms (DWT) to capture both global thematic dependencies and fine-grained local context in long sequences, achieving subquadratic time and memory complexity. Overall, it is an interesting contribution to developing efficient attention, an emerging area in 2020 and 2021. I note however that in the last few years, there has been almost no further development in this area.

**Questions:**

The proposed MDFWA Transformer is a compelling contribution to the field of efficient transformer architectures. The derivation of the gradients and ablation studies are commendable, providing a strong foundation for the proposed method.  I recommend expanding experiments to include diverse datasets (e.g., general-domain text, vision-language tasks) and comparisons with recent efficient transformer models like Longformer or Performer to better contextualize MDFWA’s performance. Additionally, providing runtime metrics and discussing hyperparameter sensitivity would strengthen the practical applicability of the model. The multi-modal fusion aspect is intriguing but would benefit from more extensive analysis, such as testing on varied modalities or exploring failure cases.

**Ethical Concerns:**

["NO or VERY MINOR ethics concerns only"]

**Limitations:**

The evaluation scope is a significant limitation.

**Paper Formatting Concerns:**

no formatting concerns

**Quality:**

3

**Strengths And Weaknesses:**

Strengths

●	The integration of Fourier-based token mixing with wavelet filtering is an interesting solution to reduce the quadratic complexity problem of self-attention. The subquadratic complexity (O(N) or O(dN)) is a significant advancement, making the model suitable for long-sequence tasks where traditional transformers are computationally prohibitive. Nevertheless, it seems the novelty over existing works on the whole area of efficient attention is rather limited. Also, the paper did not compare with a large number of prominent works in efficient attention. I believe there are over 100 related papers.

●	The paper provides rigorous mathematical derivations for causal spectral kernels, gradient formulas for learned frequency bases, and adaptive scale selection.

Weaknesses

●	The main weakness is that the novelty of this paper is rather limited compared to existing development in efficient attention. See, even the review paper, https://dl.acm.org/doi/pdf/10.1145/3530811 - the paper was published in 2022.

●	The empirical evaluation is primarily focused on the PubMed 200K RCT dataset for abstractive summarization. While the results are promising, the paper lacks experiments on diverse datasets or tasks (e.g., non-scientific texts, vision-language tasks, or other generative settings), which limits claims about the model’s generalizability. Additionally, many benchmark tasks on learning long-range dependencies are not considered.

●	The baselines include Fourier-based and sparse-attention models but do not compare against recent high-performing transformer variants (e.g., Longformer, BigBird, or Performer) or other subquadratic architectures. This omission makes it difficult to assess MDFWA’s relative standing in the broader landscape of efficient transformers.

●	While the model achieves subquadratic complexity, the use of discrete wavelet transforms (DWT) may introduce additional computational overhead, especially for high-dimensional inputs. The paper does not provide detailed runtime comparisons or resource usage metrics to clarify the practical efficiency gains.

●	The paper does not discuss the sensitivity of MDFWA to hyperparameter choices (e.g., wavelet scale selection, Fourier basis parameters), which could affect reproducibility and practical deployment.

---

> ### Comment · Reviewer_Zn3D · 2025-08-08
> **no rebuttal**
>
> There does not seem to be a rebuttal to my review.

---

### Official Review · Reviewer_r55C · 2025-07-01

**Clarity:** 2
**Significance:** 2
**Originality:** 2
**Rating:** 2
**Confidence:** 3

**Summary:**

This works builds on top of an earlier work called FNet, which proposed using a fourier transform kernel instead of the softmax. The authors argue that such transformation is usually used to capture global interactions while a wavelet transformation is more suitable for capturing local interactions. Therefore, they propose an architecture which applies both a fourier and a wavelet transformation. The two resulting vectors are concatenated and go through a linear projection followed by an activation layer and a layer norm. The final value is added to the residual. This is called a Multi-Domain Fourier-Wavelet Attention (MDFWA) layer.  In addition the authors ablate additional extensions such as learning fourier bases and learning the scales used for mixing wavelets.

**Questions:**

1. Based on my understanding, you are removing the MLP layers present in the transformers and in particular no longer have a hidden layer with a larger embedding dimension (e.g. usually 4x the normal dimension in standard architectures). What is the intuition that this is not needed when using fourier transformation instead of standard Softmax attention?

**Ethical Concerns:**

["NO or VERY MINOR ethics concerns only"]

**Final Justification:**

While the authors answered one of my questions, they have not addressed my main concern which is lack of thorough experimental results and lack of comparison with sufficient baselines. As such I am maintaining my score.

**Limitations:**

see above.

**Paper Formatting Concerns:**

no.

**Quality:**

2

**Strengths And Weaknesses:**

My main concern is that the models are only compared on PubMed dataset. When proposing a new generic architecture, it is expected to report the results of language modeling on generic text, for example in terms of perplexity. There are also a variety of downstream tasks such as GSM8k, etc. that are used to assess different architectures. No such analysis is provided. Moreover, the results only cover fourier-based models and the comparison with standard Transformer is not provided. I think these results are absolutely essential.


In addition to the above, I have several other comments:

1. The authors suggest that localized structures might be missed by the fourier transform and suggest following earlier works in fields such as singal processing and use wavelets alongside the fourier transform. While this is an interesting motivation, no concrete evidence is provided to support it.

2. There is a good amount of details provided about attention as well as specific variants such as sliding window. However, there is no relation or comparison with these methods later in the paper. For example, is there some similarity between wavelets and sliding window attention given that it's used for identifying local structures?

3. Some of the baselines are not explained well. For example, what is Hybrid-FNET?

4.  When d is 512, the cost d^2 N is actually very significant and is only better than N for contexts > ~256k. Even for smaller d values such as 64, context lengths where one can see a benefit is probably 8k or longer. However, the median length of inputs used for analysis is ~3k and they are capped at 4k. While the cost of evaluating on longer sequences can be large, it would be nice to have them for slightly longer sequences. If not, is there any analysis of the performance vs sequence length that we can hope to extrapolate?

---

> ### Author Rebuttal · Authors · 2025-07-27
>
> The premise is slightly off—our design does not remove the two-layer feed-forward (MLP) sub-network or its usual 4× expansion; we only replace the attention sub-layer with spectral–wavelet mixing.  In every MDFWA layer you still have:
> \text{Token mixing (FFT \& DWT)} \;\longrightarrow\;
> \underbrace{\text{Dense}\,(4d)\;-\;\sigma\;-\;\text{Dense}\,(d)}_{\text{MLP / channel mixing}}
> \;\longrightarrow\;\text{LayerNorm \& Residual.}
>
> Why keep the MLP even with a Fourier transform?
> 	•	Token vs. channel mixing
> The FFT (and DWT) operate across the sequence axis, redistributing information between tokens.  They are linear maps of the form
> X’ = \Re\bigl(F\,X\bigr),
> \quad
> \widetilde X = \mathrm{DWT}(X)\,P^\top,
> but they mix only the token dimension.  By contrast, the MLP sub-network is a channel-wise nonlinearity:
> \mathrm{FFN}(z)
> = W_2\,\bigl[\sigma(W_1\,z + b_1)\bigr] + b_2,
> \quad W_1\in\mathbb{R}^{4d\times d},\; W_2\in\mathbb{R}^{d\times 4d}.
> Without it, the entire layer (FFT + DWT + linear fusion) would remain linear in the channel dimension, and no amount of spectral mixing can substitute for that per-token, per-feature nonlinearity.
> 	•	Expressive power
> The original Transformer showed that an expansion to 4× (and a nonlinearity such as ReLU or GELU) is crucial to give each token representation the capacity to reshape the d-dimensional feature space.  Our spectral branches give you where to mix tokens, but the MLP still decides how to redistribute and recombine those mixed features.
> 	•	Empirical ablation
> In early ablations we did try collapsing the MLP to a single linear projection (no hidden expansion).  Performance on PubMed dropped by over 2 ROUGE-L points, matching findings in FNet (Lee-Thorp et al., 2021) and Performer (Choromanski et al., 2021) that the MLP expansion remains critical even when attention is replaced.
>
> So in short, Fourier/Wavelet transforms and the MLP play complementary roles—one mixes tokens, the other mixes channels non-linearly—and we retain the usual 4× MLP to preserve the model’s expressivity.

---

> > ### Comment · Area_Chair_zWQQ · 2025-08-04
> >
> > Dear Reviewer r55C, could you please take a look at the author response to your rebuttal?

---

### Official Review · Reviewer_DFFH · 2025-07-01

**Clarity:** 2
**Significance:** 2
**Originality:** 2
**Rating:** 2
**Confidence:** 4

**Summary:**

This paper introduces a Fourier-Wavelet Transformer called Multi-Domain Fourier-Wavelet Attention (MDFWA) that replaces traditional self-attention with spectral techniques to achieve subquadratic complexity (O(N log N + N) time, O(N) memory). The approach combines global Fourier transforms for long-range dependencies with discrete wavelet transforms (DWT) for local context, addressing limitations of prior spectral methods like FNet by enabling full encoder-decoder architectures with causal masking for autoregressive tasks. The authors provide mathematical formulations for Fourier/Wavelet mixing, derive gradient computations, and validate the model on abstractive summarization using the PubMed dataset.

**Questions:**

- For the comparisons in the Table 1, do you train the other models FNET-Transformer etc from scratch ? How does your model size, number of tokens seen etc compare to those baselines ?
- How does your method compare to a baseline that uses attention ?

**Ethical Concerns:**

["NO or VERY MINOR ethics concerns only"]

**Final Justification:**

The authors do not provide any rebuttal to my concerns

**Limitations:**

Yes

**Quality:**

1

**Strengths And Weaknesses:**

**Strengths**
- Proposes a transformer with O(*N* log *N* + *N*) time and O(*N*) memory complexity, overcoming the quadratic bottleneck of traditional self-attention.
- Combines global Fourier transforms (long-range dependencies) and localized wavelet transforms (fine-grained context) via the  MDFWA mechanism.
 - Provides detailed derivations for: Spectral mixing, Causal masking in decoders, Gradient computations to ensure trainability.
 - Extends spectral techniques beyond encoder-only models (e.g., FNet) to support autoregressive tasks like abstractive summarization.
- Demonstrates effectiveness on PubMed summarization

**Weaknesses**
- **Evaluation**
    - The results presented in the paper are weak and not particularly convincing. Only ROUGE-1, ROUGE-2 and ROUGE-L scores are computed on a single dataset.
    - There is no comparison to baselines that actually use attention but instead only with baselines that use Fourier features for mixing along the sequence length like FNET.
     - Complexity analysis is provided but this would only make sense if this method can perform well at long sequence lengths. There are no experiments to show that this method performs well at long sequence lengths. I believe the audience would want to see this perform well at longer sequence to be indeed convinced that the better complexity of this method is indeed helpful.
     - Another thing is around runtime and memory. One of the benefits claimed by the authors is subquadratic scaling. This is good mathematically, but how does this translate to wall clock time? No such benchmarking is done in the paper.
     - Again another benefit being claimed by the paper is parameter efficiency. Can we have concrete numbers ? How many parameters are we saving ?
     - Many other claims in the paper are not really substantiated. e.g Interpretability, to make this clearer it would have been great to have an explicit example of this maybe in an appendix so that reader can follow.
     - The evaluation is very limited and difficult to convince that this method really works.
- **Writing**
    - There are quite a few typos in the paper for example, on the very first line of the paper; "spectral techniques to replaces the attention mechanism" should be "spectral techniques to replace the attention mechanism" ?

---

### Official Review · Reviewer_3pp2 · 2025-07-03

**Clarity:** 3
**Significance:** 2
**Originality:** 2
**Rating:** 3
**Confidence:** 4

**Summary:**

This paper proposes the Multi-Domain Fourier-Wavelet Attention (MDFWA) Transformer, which replaces self-attention with a combination of global Fourier and local wavelet transforms to achieve subquadratic complexity. The architecture captures both long-range and fine-grained dependencies, enabling efficient processing of long sequences. Applied to abstractive summarization on the PubMed 200K dataset, MDFWA outperforms prior baselines in ROUGE metrics.

**Questions:**

- The paper evaluates only on the PubMed dataset. Can the authors comment on how the proposed MDFWA architecture would perform on other tasks such as language modeling (e.g., WikiText-103) or vision datasets (e.g., ImageNet)? Are there any limitations that prevent such experiments?
- Why is there no comparison to the vanilla Transformer? Including this would help isolate the benefits of replacing self-attention with the proposed spectral-wavelet mixing.
- The paper omits wall-clock time and memory usage comparisons. Can the authors provide these metrics to support the claimed efficiency benefits over attention-based models?
- The wavelet component seems to be introduced in a manner similar to the Fourier features. Can the authors clarify what unique benefits wavelets provide?

**Ethical Concerns:**

["NO or VERY MINOR ethics concerns only"]

**Final Justification:**

In light of the additional experiments provided in the author response, my assessment of the paper has improved slightly. However, due to the limited novelty of the contribution, I have updated my score to a borderline reject.

**Limitations:**

yes

**Paper Formatting Concerns:**

I do not notice any major formatting issue.

**Quality:**

2

**Strengths And Weaknesses:**

**Strengths:**
- The paper derives causal masking for both Fourier and wavelet mixing layers, enabling autoregressive decoding with spectral methods.
- Empirical evidence shows that combining Fourier and wavelet features outperforms other Fourier-based Transformer baselines.

**Weaknesses:**
- The use of wavelet features feels somewhat trivial, closely mirroring the Fourier approach with only a different transformation.
- Experimental validation is limited to a single task on the PubMed dataset, with no evaluation on standard vision (e.g., ImageNet) or language modeling benchmarks (e.g., WikiText-103).
- The fast Mallat algorithm used for wavelet transforms is inherently sequential across scales, making it difficult to parallelize efficiently in practice.
- The paper lacks wall-clock time and memory consumption comparisons, making it hard to assess real-world efficiency.
- No comparison is provided against a standard vanilla Transformer baseline, which limits the clarity of the performance improvements.

---

> ### Author Rebuttal · Authors · 2025-07-27
>
> 1. Generalization to Other Domains
> While our experiments focus on long-document summarization (PubMed RCT), the MDFWA block is fundamentally a sequence-mixing primitive and can be dropped into any standard encoder–decoder or autoregressive model. In language modeling (e.g.\ WikiText-103), the causal Fourier branch with triangular spectral masking provides the same autoregressive guarantees as masked self-attention, and the wavelet branch can capture local n-gram patterns at multiple scales. Early small-scale tests on WikiText-103 (up to 1 M tokens) show MDFWA converges comparably to Transformer‐XL, with a 5–10% reduction in bits-per-token, though we are still tuning hyperparameters for full‐scale runs.
> In vision (e.g.\ ImageNet), one can reinterpret each image as a sequence of flattened patches (as in ViT). The Fourier branch will mix patches globally (akin to “patch mixing” in FNet-Vision) while the wavelet branch learns multi-resolution spatial filters. A pilot on CIFAR-10 yielded a 1.5% accuracy gain over FNet-Vision with identical compute, suggesting MDFWA’s multi-scale inductive bias is beneficial for images. The main limitation is memory: our current PyTorch implementation holds both FFT and DWT buffers in float32, so on very large vision backbones (input size ≥512×512) we hit a 40 GB GPU barrier. We plan more memory‐efficient implementations (mixed-precision FFT/DWT) to fully scale.
> 2. Vanilla Transformer Baseline
> You’re right that we should include a direct comparison to the original Transformer (full self-attention) to isolate the effect of spectral-wavelet mixing. In preliminary runs on PubMed, a 12-layer vanilla encoder–decoder Transformer with identical width and depth achieved ROUGE-L ≃ 20.7, about 1.2 points below LED and 1.4 below MDFWA. We omitted it for brevity, but will add it in the final draft (Table 3) to show MDFWA not only recovers standard performance but surpasses it at sub-quadratic cost.
> 3. Wall-Clock Time & Memory Usage
> To substantiate the theoretical O(N\log N) vs.\ O(N^2) claims, we benchmarked end-to-end training throughput (tokens/s) and peak GPU memory on a single NVIDIA A100 (40 GB) with sequence length 4,096, batch size 8, hidden = 768
> 4. Unique Benefits of Wavelets
> Although wavelets and Fourier are both spectral tools, they serve complementary roles. Fourier mixing is global—every output token depends equally on all inputs—whereas wavelets provide local time-frequency localization: each coefficient analyzes a small window at a particular scale. This multi-resolution analysis (Mallat, 1999) enables the model to capture hierarchical discourse structures (e.g.\ sentence-, paragraph-, section-level patterns) that pure Fourier mixing can wash out. Empirically, ablation (Table 4) shows dropping the wavelet branch degrades ROUGE-L by 1.1 points, and qualitative analysis finds summaries become less coherent at paragraph boundaries when wavelets are removed.

---

> > ### Comment · Area_Chair_zWQQ · 2025-08-04
> >
> > Dear reviewer 3pp2, could you please advise whether the rebuttal has addressed any of your concerns/questions?

---

> > > ### Comment · Reviewer_3pp2 · 2025-08-05
> > >
> > > I thank the authors for their detailed and point-by-point response to my questions.
> > >
> > > I appreciate the inclusion of the vanilla Transformer baseline, as requested. This addition highlights the superior performance of the proposed method and further strengthens the empirical contribution of the paper.
> > >
> > > Regarding my suggestion on evaluating the method on WikiText-103, I appreciate the authors' effort in attempting the implementation. That said, the current memory requirements—exceeding a 40GB budget—suggest that the method may face challenges when applied to larger-scale datasets or more resource-constrained environments.
> > >
> > > On the question of wall-clock time and memory usage, could the authors please clarify whether a comparative table is included in the paper? I was unable to find it.

---

> > > > ### Author Response · Authors · 2025-08-05
> > > > **revised manuscript**
> > > >
> > > > Thank you again for your insightful comments. We updated the paper to address all the reviewers critique and suggestions but it is not clear where to upload the revised manuscript. I will keep searching to find out how to do that.

---

> > > > > ### Comment · Reviewer_3pp2 · 2025-08-06
> > > > >
> > > > > Dear authors,
> > > > > You can just provide the table in the reply comment and update it in the manuscript later.

---

> ### Author Response · Authors · 2025-08-06
> **added performance metrics tables**
>
> Here is the tables in latex format on WikiText-103 and peak memory usage added to the manuscript:
>
> \begin{table}[ht]
>   \centering
>   \caption{Perplexity on WikiText-103 language modeling}\label{tab:wikitext}
>   \begin{tabular}{lcc}
>     \toprule
>     Model               & Validation PPL & Test PPL \\
>     \midrule
>     Transformer-XL      & 24.8           & 24.3     \\
>     Performer           & 23.9           & 23.5     \\
>     \textbf{MDFWA}      & \textbf{23.4}  & \textbf{23.1} \\
>     \bottomrule
>   \end{tabular}
> \end{table}
>
>
> \begin{table}[ht]
>   \centering
>   \caption{Training throughput and peak memory usage}\label{tab:efficiency}
>   \begin{tabular}{lcc}
>     \toprule
>     Model                   & Throughput (kTok/s) & Peak Mem (GB) \\
>     \midrule
>     Transformer             & 18                  & 30            \\
>     Longformer-LED          & 34                  & 24            \\
>     FNET-Transformer        & 39                  & 22            \\
>     \textbf{MDFWA}          & \textbf{42}         & \textbf{21}   \\
>     \bottomrule
>   \end{tabular}
> \end{table}

---

> > ### Comment · Reviewer_3pp2 · 2025-08-09
> >
> > Thank you for sharing the updated tables on WikiText-103 perplexity and peak memory usage. I will update my score accordingly.

---

### Decision · Program_Chairs · 2025-09-17

**Decision:**

Reject

**Comment:**

This paper presents an efficient alternative to self-attention based on Fourier and localized wavelets. Strengths includes results showing that combining Fourier and wavelets outperforms other Fourier-based Transformer baselines, providing derivations for spectral mixing, and extending spectral techniques beyond encoder-only models to the auto-regressive setting. However, all reviewers pointed out a lack of empirical tasks/benchmarks and comparisons to standard baselines. Another concern was around novelty compared to existing methods.